# From Bits to Chips: An LLM-based Hardware-Aware Quantization Agent for Streamlined Deployment of LLMs

## Abstract

Deploying models, especially large language models (LLMs), is becoming increasingly attractive to a broader user base, including those without specialized expertise. However, due to the resource constraints of certain hardware, maintaining high accuracy with larger model while meeting the hardware requirements remains a significant challenge. Model quantization technique helps mitigate memory and compute bottlenecks, yet the added complexities of tuning and deploying quantized models further exacerbates these challenges, making the process unfriendly to most of the users. We introduce the Hardware-Aware Quantization Agent (HAQA), an automated framework that leverages LLMs to streamline the entire quantization and deployment process by enabling efficient hyperparameter tuning and hardware configuration, thereby simultaneously improving deployment quality and ease of use for a broad range of users. Our results demonstrate up to a $2.3\times$ speedup in inference, along with increased throughput and improved accuracy compared to unoptimized models on Llama. Additionally, HAQA is designed to implement adaptive quantization strategies across diverse hardware platforms, as it automatically finds optimal settings even when they appear counterintuitive, thereby reducing extensive manual effort and demonstrating superior adaptability. Code will be released.

## 1 Introduction

From traditional machine learning models to the recently acclaimed large language models (LLMs), deep learning technology has emerged as a pivotal force driving intelligent transformation across industries LeCun et al. (2015); Brown et al. (2020); Schmidhuber (2015). However, deploying these technologies poses two key challenges: maintaining high accuracy and reducing computational costs during deployment Sze et al. (2017); Han et al. (2015). High accuracy ensures reliable performance in practical scenarios, while low computational cost dictates deployment feasibility, particularly on resource-constrained edge devices with limited memory and processing power. One of the most effective solutions to this challenge is model quantization Jacob et al. (2018), which significantly reduces memory usage while preserving model accuracy. However, quantized models, particularly large language models, require different hyperparameter configurations during fine-tuning compared to full-precision models. Furthermore, deploying these models on specific hardware platforms necessitates additional hyperparameter adjustments to ensure compatibility and optimal performance Sze et al. (2017); Lin et al. (2019), making the process even more complex and time-consuming. As a result, these tasks are especially challenging for non-expert practitioners seeking to effectively utilize quantized models (shown in Figure 1 (a)).

Traditional hyperparameter optimization methods, such as random search Bergstra & Bengio (2012) and Bayesian optimization Snoek et al. (2012), have been used for fine-tuning models but face limitations in high-dimensional spaces, where random and grid searches are inefficient and Bayesian optimization struggles with the computational complexity of surrogate models like Gaussian Processes. On the other hand, deploying fine-tuned models on target hardware platforms poses distinct challenges, particularly in implementing hardware-specific optimizations to accelerate inference. Compiler-based methods Ragan-Kelley et al. (2013) use heuristics and predefined rules to guide optimization but are insufficient for the large search spaces required for LLMs. Machine-learning-based approaches Chen

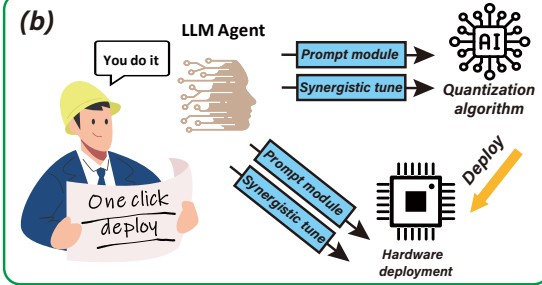

**Figure 1:** Comparison of human-driven (a) and agent-based (b) optimization for AI model deployment. Deploying an LLM on-device usually involves quantizing the model to lower bit precision to reduce memory consumption and tailoring deployment strategies for each hardware platform—steps that require repeated expert tuning and still may fail to deliver optimal performance. With more non-expert users wanting to run LLMs locally, manual adjustment of quantization parameters and deployment configurations becomes impractical. We propose an LLM-based quantization agent to automate and streamline the entire workflow, enabling efficient hyperparameter search and hardware-specific configuration. Unlike experts who tweak one parameter at a time, our agent jointly tunes all settings for synergistic optimization and consistent performance.

et al. (2018) improve upon this by training models for hardware-specific tuning but demand extensive program execution data, leading to high computational costs.

To further advance the hyperparameter optimization process, LLM-based agents Zhang et al. (2023b); Liu et al. (2024); Huang et al. (2023) leverage the superior reasoning capabilities of LLMs to serve as intelligent assistants that guide the process. LLM agents automate complex processes by interpreting high-level instructions, reducing the need for domain-specific expertise while dynamically adapting to complex optimization tasks. Unlike traditional methods, this approach potentially eliminates the reliance on exhaustive search Hong et al. (2023); Yao et al. (2022) or hardware-specific heuristics, significantly lowering the computational and expertise barriers. However, to the best of our knowledge, no existing work has introduced an agent capable of performing hyperparameter optimization specifically tailored for quantization and hardware deployment.

While LLM agents have been explored in various domains, this work is the *first* to utilize LLM agents for optimizing hyperparameters during both the fine-tuning and deployment of quantized models, as illustrated in Figure 1 (b). In this work, we propose the **H**ardware-**A**ware **Q**uantization **A**gent (**HAQA**), which provides optimized configurations for model quantization fine-tuning (e.g., learning rate, batch size, etc.) and hardware deployment (e.g., bit-width selection, kernel tuning, etc.). HAQA is a dynamic and automated framework that adjusts these configurations based on the model's accuracy and latency, ensuring compatibility and optimal performance across various scenarios. HAQA has standardized workflow and user-friendly design, thus promoting more efficient and accessible deployment strategies. We conduct extensive experiments on both computer vision models and LLMs, and deploy the quantization- and deployment-optimized model on different hardware platforms. The results demonstrate that HAQA consistently achieves higher accuracy compared to state-of-the-art quantization algorithms, while significantly reducing inference latency and improving computational throughput on different hardware platforms. On LLaMA2-7B, HAQA achieves $2.3\times$ speedup for inference while also improves accuracy compared to traditional quantization method. Moreover, we design HAQA to be hardware-adaptive regarding hardware configurations, as it automatically find correct settings even when they appear counterintuitive (e.g., 8-bit is faster than 4-bit on the Oneplus 11 mobile device equipped with Snapdragon 8 Gen 2 SoC and Qualcomm Adreno 740 GPU). The key contributions of our work are summarized as follows:

- This work introduces HAQA, a novel framework that optimizes both the fine-tuning parameters of quantized models and the hyperparameters for deployment on specific hardware platforms. To the best of our knowledge, this is the *first time* that an LLM-based agent successfully optimizes both quantization and deployment hyperparameters.

- HAQA streamlines the quantization and deployment process through a standardized and practical hyperparameter optimization workflow. This approach not only enhances the efficiency of fine-tuning and deployment but also reduces the barrier for practitioners, enabling non-experts to easily deploy and fine-tune quantized models.

- We design HAQA to be hardware-adaptive by thoroughly analyzing hardware configurations to identify subtle yet complex issues that may go unnoticed by humans and applying appropriate settings to reduce manual effort. Experimental results demonstrate that HAQA achieves significant performance improvements and exhibits superior adaptability across different hardware types.

## 2 RELATED WORK

### 2.1 LLM AGENT

With the rapid evolution of LLMs, their knowledge, reasoning, planning, and generalization have significantly advanced real-world applications Achiam et al. (2023); Hong et al. (2023); Wang et al. (2023); Park et al. (2023); Yao et al. (2022). LLM agents have also shown potential in automating machine learning workflows. MLAgentBench Huang et al. (2023) provides a benchmark for evaluating LLM agents. MLCopilot Zhang et al. (2023c) leverages LLMs to analyze historical experiments, offering interpretable solutions for novel tasks with minimal trials. AutoML-GPT Zhang et al. (2023a) automates the pipeline by dynamically generating prompts. AgentHPO Liu et al. (2024) and LLM Agent Zhang et al. (2023b) autonomously optimize hyperparameters more effectively than traditional methods. Yang et al. (2023) employs an LLM agent as optimizer, and Liu et al. (2024) uses an agent for traditional model optimization. However, these agents mainly focus on hyperparameter optimization, lacking support for tasks like quantized model training or adapting to hardware constraints.

### 2.2 MODEL QUANTIZATION

Model quantization is a key technique for optimizing deep learning models Krishnamoorthi (2018); Jacob et al. (2018), especially for deployment on resource-constrained devices. Quantization methods are generally divided into two approaches: Quantization-Aware Training (QAT) Dong et al. (2019); Zhou et al. (2016); Nagel et al. (2019) and Post-Training Quantization (PTQ) Liu et al. (2021); Nagel et al. (2020). However, QAT methods are difficult to scale to large models like LLMs, while directly applying PTQ often fails to deliver satisfactory LLM performance. GPTQ Frantar et al. (2022) proposes a PTQ method leveraging second-order error compensation. AWQ Lin et al. (2024) improves PTQ by introducing activation-aware weight quantization to prioritize critical weights. OWQ Lee et al. (2024) further refines this with an outlier-aware mixed-precision scheme, retaining sensitive weights in higher precision. QLoRA Dettmers et al. (2024) combines low-bit quantization with low-rank adapters Hu et al. (2021).

### 2.3 HARDWARE OPTIMIZATION

Optimizing LLM inference across diverse hardware platforms has been extensively studied. LUT-GEMM Park et al. (2022) proposes customized matrix multiplication kernels to boost efficiency in quantized models, while Atom Zhao et al. (2024) employs 4-bit quantization with mixed-precision weight reordering to improve adaptability and performance. Another line of work supports neural networks on general hardware platforms using code generation. Halide Ragan-Kelley et al. (2013) decouples computation from scheduling for flexible tuning, while TVM Chen et al. (2018) applies graph- and operator-level optimizations for diverse hardware backends. Additionally, Cowan et al. (2020) designs efficient layouts for quantized weights and auto-tuners to execute layers efficiently. These methods automate hardware-aware optimizations but depend heavily on cost models or black-box auto-tuners requiring extensive training data and time. This work leverages LLM agents to adaptively optimize quantization, addressing hardware-specific and model-level constraints while bridging the gap with intelligent agents.

## 3 METHODOLOGY

A unified optimization process is crucial for both fine-tuning and deployment. This section introduces HAQA, a comprehensive framework that jointly optimize the hyperparameter of quantization model fine-tuning and hardware deployment, ensuring high accuracy and inference speed. We organize this section as follows: Section 3.1 proposes the HAQA **Prompt Design** that comprises the comprehensive

**(a)** I am running the model on an Ubuntu machine equipped with an NVIDIA A6000 GPU fully supporting TensorRT. Here's more details about the hardware:{xx}. *Memory Limit* is 10 GB.

Note that there are *x rounds left*, please try to make effective attempts
The current configuration is {...}, Evaluation Result:{Task1': 'acc'; ... }
List of training losses:[]
*Please check the history and think about your next plan before action.*
*History:* [{"round": x-1; "user_message": a; "Agent_response": b},...]

The current execution configuration is {...}, and the deployment result is: {"Kernel":"Softmax", "latency":...}. The whole model latency is: [].
*Please check the history and think about your next plan before action.*
**(d)** *History:* [{"round": x-1, "user_message": a, "Agent_response": b}, ...]

**(b)** *I am deploying* the llama3.2-3b model. The LLaMA model consists of various CUDA kernels. The specific kernel information, default execution configuration, and core kernel code will be provided. The deployment latency results will be fed back to you.
*Here is the search space:*{ "kernel":... , "input size":..., ...}
*Core Code for deployment :* b.py, ...

*I also need to fine-tune* LLaMA2-13b model using QLoRA. The specific core code and the configuration will be provided. The dataset is alpaca. There are multiple validation datasets, and the results will be fed back to you
*Here is the search space:*{ "learning_rate": ...}
*Core Code for finetuning :* a.py, ...
**(c)**

**Figure 2:** This Figure illustrates a sample prompt, demonstrating their practical structure. Blue text highlights key information conveyed to the agent. Purple text is used to enhance agent's decision-making ability, which we'll discuss in Section 3.3. The full prompt and detailed descriptions of each component is in Appendix E. Panels (a)–(c) denote the static prompt's components: (a) hardware platform information, (b) deployment objective details , and (c) fine-tuning objective details, while panel (d) represents the dynamic prompt, which contains logs and conversation history.

Static and Dynamic Prompts in optimization. Section 3.2 defines the **LLM Agent Workflow** of HAQA for iterative, efficient optimization. Section 3.3 demonstrates the **User-Friendliness** of HAQA that ensures an adaptable and efficient user experience. Section 3.4 introduces the **Adaptive Quantization Strategies** of HAQA, which dynamically selects optimal quantization configurations based on hardware-specific attributes.

## 3.1 HAQA PROMPT DESIGN

We categorize the prompt into two components: *Static Prompt* and *Dynamic Prompt*, as shown in Figure 2, enabling a modular design that enhances optimization efficiency while improving usability by simplifying prompt modification and reuse.

The *Static Prompt* encapsulates foundational information that remains largely unchanged throughout the optimization process. It first clearly describes the hardware platform specification, followed by comprehensive optimization objectives regarding the model, quantization strategy, and hardware platform (shown in Figure 2 (a) (b) (c)). These details include the core code of the fine-tuning framework and model, deployment environment, hardware optimization strategies, and the hyperparameter search space, which may vary significantly across tasks. Specifically, we adopt a kernel-wise optimization strategy, decomposing the model into individual computational kernels. The static prompt thus includes essential kernel-specific details, such as kernel names, input dimensions, and quantization configurations (shown in Figure 2 (b)). Standardizing the Static Prompt effectively mitigates inconsistencies in task descriptions caused by frequent modifications.

On the other hand, the *Dynamic Prompt* is designed for iterative updates, incorporating real-time optimization metrics such as fine-tuning parameter configurations, task-specific accuracy, and latency (shown in Figure 2 (d)). It also includes kernel execution parameters, structured into two key components. The first addresses kernel-specific optimizations, including computation block size for parallelization, tiling size for efficient memory access, and loop unrolling to reduce control overhead and enhance performance. The second defines execution strategies, specifying memory hierarchy configurations for optimized tensor placement (e.g., global, shared, or local memory) and thread scheduling policies to balance workload distribution.

## 3.2 HAQA WORKFLOW

As shown in Figure 3, the Dynamic Prompt combines with Static Prompt and forms a comprehensive prompt sent to the agent for further optimization. After fine-tuning the quantization model and deploying it for inference speed test, the accuracy and speed data are incorporated into the Dynamic Prompt for the next iteration (we show the updated Dynamic Prompt in Appendix E). HAQA then refines the optimization guidance based on the feedback and initiates a new experiment. The process continues iteratively until the objectives are met or the maximum number of iterations is reached.

During the experiments, we identified several issues with the responses of HAQA:

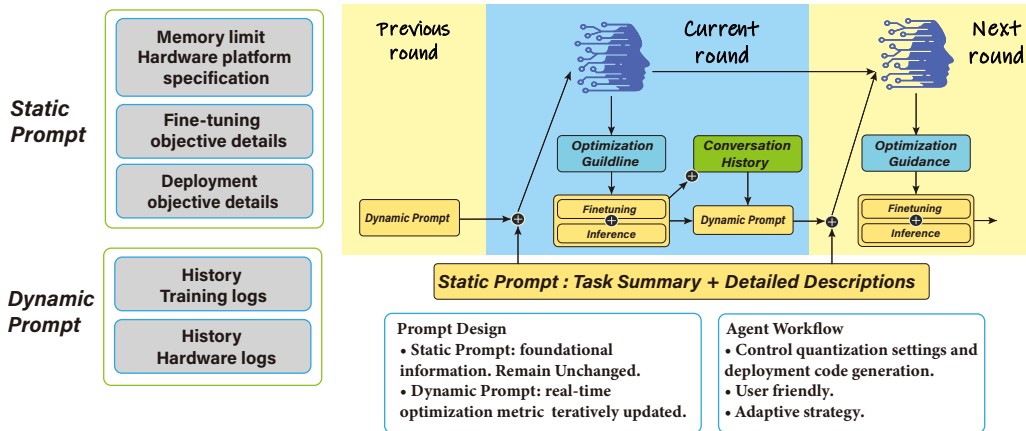

**Figure 3:** To enable the LLM-based agent, we first design a prompt with two parts: the static and dynamic prompts. The static prompt includes configurations that remain unchanged, such as memory limits, platform specifications, hardware details, and quantization algorithm hyperparameters. The dynamic prompt contains information that changes dynamically, such as hardware speed testing logs and quantization result logs. During the quantization and deployment stage, the agent generates optimization guidelines based on the current prompt. The quantization algorithm and hardware deployment then execute based on the guidelines, providing accuracy and speed feedback. Using this feedback and the prompt history, the agent updates the dynamic part of the prompt and generates a new guideline.

1. Some responses did not adhere to the required format.
2. Certain configurations violated predefined constraints.
3. Some responses contained irrelevant information unrelated to the task.

To address these issues, we incorporate ReAct Yao et al. (2022) into the prompt (as highlighted in purple in Figure 2(d) and Appendix E). ReAct is a method of prompt design which combines reasoning and action. Specifically, ReAct designs prompts illustrating the full process—from defining task goals and internal reasoning, to taking actions, receiving feedback, and generating final answers. By integrating ReAct, our framework curtails blind exploration, boosts optimization efficiency and stability, enhances interpretability, and supports smarter, more effective decision-making.

### 3.3 USER-FRIENDLINESS OF HAQA

The proposed framework prioritizes user-friendliness to ensure a seamless and efficient user experience. First, HAQA generates task logs at the end of each task, providing users with a clear record of configurations, results, and optimization progress. Second, HAQA allows users to control the length of the optimization history based on their preferences or task requirements. During experiments, we observed that if the history length is not properly managed, it may exceed the maximum input length of agent, leading to workflow interruptions. By dynamically controlling the length of the history, the framework avoids such interruptions and ensures smooth optimization. Additionally, HAQA provides optimization suggestions based on the current context and experimental results. Users can refine the their own prompt and adjust experimental settings according to the agent's suggestions and their specific needs, enhancing the flexibility and customizability of the system.

### 3.4 ADAPTIVE QUANTIZATION STRATEGIES DESIGN

HAQA is designed to implement adaptive quantization strategies across diverse hardware platforms. Due to architectural variations (i.e., cache size, memory bandwidth) and differing levels of instruction set support, model inference performance under various quantization configurations can fluctuate significantly depending on the hardware. For example, while INT4 quantization may yield superior performance over INT8 on an NVIDIA A6000, the opposite may hold true for mobile devices. Additional details and experimental results are provided in Section 4.4. This variability complicates model deployment, particularly for developers lacking expertise in hardware-specific optimizations, potentially resulting in suboptimal performance. On the other hand, HAQA integrates hardware analysis capabilities, allowing it to dynamically select and apply the most suitable quantization

**Table 1:** Accuracy (%) for ResNet models under different quantization bit-widths with various hyperparameter optimization methods.

| Model | Precision | Default | Human | Local search | Bayesian opt. | Random search | NSGA2 | HAQA |
|---|---|---|---|---|---|---|---|---|
| ResNet20 | w8a8 | $89.87 \pm 0.50$ | $92.25 \pm 0.28$ | $91.60 \pm 0.38$ | $91.69 \pm 0.42$ | $91.58 \pm 0.40$ | $91.64 \pm 0.35$ | $92.80 \pm 0.22$ |
| | w4a4 | $81.15 \pm 0.46$ | $87.17 \pm 0.42$ | $82.88 \pm 0.32$ | $82.29 \pm 0.28$ | $82.16 \pm 0.30$ | $81.65 \pm 0.36$ | $88.38 \pm 0.10$ |
| | w2a2 | — | $74.23 \pm 0.38$ | $74.93 \pm 0.41$ | $74.96 \pm 0.37$ | $70.57 \pm 0.33$ | $73.16 \pm 0.39$ | $75.55 \pm 0.32$ |
| ResNet32 | w8a8 | $92.94 \pm 0.44$ | $94.52 \pm 0.33$ | $94.67 \pm 0.31$ | $94.76 \pm 0.29$ | $94.65 \pm 0.34$ | $94.71 \pm 0.37$ | $94.98 \pm 0.19$ |
| | w4a4 | $73.40 \pm 0.52$ | $85.33 \pm 0.47$ | $88.90 \pm 0.30$ | $88.31 \pm 0.33$ | $88.18 \pm 0.36$ | $86.52 \pm 0.38$ | $90.02 \pm 0.21$ |
| | w2a2 | — | $84.24 \pm 0.39$ | $87.23 \pm 0.40$ | $87.26 \pm 0.35$ | $82.87 \pm 0.32$ | $85.46 \pm 0.39$ | $87.85 \pm 0.38$ |
| ResNet50 | w8a8 | $74.30 \pm 0.53$ | $74.31 \pm 0.35$ | $74.80 \pm 0.34$ | $74.83 \pm 0.30$ | $74.79 \pm 0.36$ | $74.81 \pm 0.33$ | $74.89 \pm 0.27$ |
| | w4a4 | $71.40 \pm 0.48$ | $71.97 \pm 0.46$ | $71.91 \pm 0.29$ | $71.74 \pm 0.27$ | $71.70 \pm 0.31$ | $71.60 \pm 0.34$ | $72.24 \pm 0.16$ |
| | w2a2 | — | $67.10 \pm 0.41$ | $67.98 \pm 0.42$ | $68.00 \pm 0.38$ | $64.58 \pm 0.30$ | $66.60 \pm 0.40$ | $68.46 \pm 0.25$ |

strategies for a given platform. By leveraging hardware-specific attributes—such as architecture, computational power, and supported instruction sets—HAQA determines the optimal quantization configurations, maximizing performance while adhering to hardware constraints.

# 4 EXPERIMENT

The evaluation systematically assesses the effectiveness of the proposed method. First, in Section 4.2 we present the accuracy results of quantized models across multiple datasets, highlighting the performance improvements achieved by HAQA. Second, in Section 4.3 we analyze deployment outcomes, demonstrating HAQA's ability to accelerate inference under specific hardware constraints. Finally, in Section 4.4, we investigate non-intuitive configurations identified during optimization, showcasing HAQA's capacity to discover solutions that extend beyond conventional approaches. We also present the hyperparameter search space in Appendix C and analyze the cost and overhead of HAQA in Appendix D.

## 4.1 EXPERIMENTAL SETUPS

**Fine-tuning:** We conduct experiments using a diverse set of models, including ResNet20, ResNet32, ResNet50, LLaMA2-7B, LLaMA2-13B, LLaMA3.2-3B, and LLaMA3-8B. Specifically, ResNet20 and ResNet32 are trained on CIFAR-10, ResNet50 on ImageNet2012, and the LLaMA models on the Alpaca dataset (tatsu-lab/alpaca). For model quantization, we employ DoReFa and QLoRA, both of which have demonstrated superior performance on traditional vision models and large-scale language models. The initial hyperparameters for fine-tuning are derived from Choi et al. (2018); Zhou et al. (2016); Geng & Liu (2023); ds3001 (2025), ensuring a well-calibrated starting point for optimization. All experiments are conducted using ChatGPT-4 Achiam et al. (2023) (specifically GPT-4-0613) as agent.

**Deployment:** For hardware deployment, our system is built upon llama.cpp contributors (2023), a widely used and efficient framework for end-to-end LLM inference across diverse hardware architectures. We deploy LLaMA2-7B, LLaMA2-13B, LLaMA3.2-3B, and LLaMA3-8B under three quantization types: FP16, INT8, and INT4. To evaluate inference efficiency on hardware platforms, we conduct experiments on an NVIDIA A6000 GPU with 48 GB memory using CUDA 12.3. The primary metric for evaluation is token generation time, with an input sequence length of 128 and output token length of 256. To assess the effectiveness of HAQA's adaptive quantization strategies, we also run experiments on a OnePlus 11 mobile device powered by Snapdragon 8 Gen 2 SoC, featuring an octa-core Kryo CPU and an Adreno 740 GPU. On this device, we deploy three models: openllama-3B, tinyllama-1.1B, and gpt2-large-774MB. Each experiment is repeated 10 times, and the average result is taken as the final metric.

## 4.2 CAPABILITY TO IMPROVE ACCURACY ACROSS DIFFERENT MODELS

We first conduct experiments to evaluate the performance of ResNet using DoReFa QAT Zhou et al. (2016). The experiments cover various quantization bit-widths and hyperparameter optimization methods, as summarized in Table 1. In the table, "Default" denotes hyperparameters optimized for full-precision models, while "Human" refers to the average results from manual tuning by experienced

**Table 2:** Accuracy (%) comparison of different LLaMA models across various tasks and hyperparameter optimization methods. HAQA consistently outperforms other methods.

| Model | Precision | Method | BoolQ | RTE | Winogrande | OpenBookQA | ARC-C | ARC-E | Hellaswag | MathQA | AVG |
|---|---|---|---|---|---|---|---|---|---|---|---|
| LLaMA2-7B | INT4 | Human | 75.68 | 67.51 | 69.43 | 35.04 | 47.26 | 77.30 | 51.56 | 38.90 | 62.04 |
| | | Local | 76.46 | 67.15 | 68.95 | 35.60 | 46.48 | 76.86 | 51.26 | 38.63 | 61.99 |
| | | Bayesian | 74.90 | 68.16 | 68.07 | 35.40 | 44.95 | 76.65 | 51.37 | 37.88 | 61.49 |
| | | Random | 73.34 | 67.96 | 68.65 | 35.20 | 44.52 | 76.26 | 51.46 | 38.37 | 61.29 |
| | | NSGA2 | 71.09 | 66.43 | 68.36 | 35.40 | 44.14 | 76.46 | 51.65 | 38.27 | 60.79 |
| | | **HAQA** | 77.25 | 68.60 | 69.53 | 37.00 | 48.24 | 78.02 | 51.86 | 39.84 | 63.11 |
| | INT8 | Human | 77.15 | 64.26 | 68.26 | 35.40 | 47.26 | 77.30 | 51.60 | 39.40 | 61.94 |
| | | Local | 79.13 | 68.88 | 69.46 | 37.67 | 48.03 | 78.01 | 51.44 | 39.17 | 62.90 |
| | | Bayesian | 77.37 | 69.69 | 68.68 | 37.22 | 46.50 | 78.05 | 51.30 | 38.67 | 62.36 |
| | | Random | 75.71 | 69.89 | 69.11 | 37.17 | 46.12 | 77.46 | 51.54 | 38.91 | 62.17 |
| | | NSGA2 | 73.51 | 68.11 | 69.02 | 37.17 | 45.64 | 77.76 | 51.78 | 39.21 | 61.70 |
| | | **HAQA** | 80.30 | 71.48 | 70.24 | 38.60 | 49.70 | 79.00 | 51.90 | 41.13 | 64.22 |
| LLaMA2-13B | INT4 | Human | 79.50 | 67.10 | 71.97 | 36.00 | 52.24 | 78.41 | 52.05 | 40.40 | 64.20 |
| | | Local | 80.31 | 67.14 | 71.49 | 35.80 | 52.15 | 79.21 | 51.82 | 40.16 | 64.42 |
| | | Bayesian | 78.75 | 68.15 | 70.61 | 35.60 | 50.62 | 79.00 | 51.93 | 39.41 | 63.92 |
| | | Random | 77.19 | 67.95 | 71.19 | 35.40 | 50.19 | 78.61 | 52.02 | 39.90 | 63.72 |
| | | NSGA2 | 74.94 | 66.42 | 70.90 | 35.60 | 49.81 | 78.81 | 52.21 | 39.80 | 63.22 |
| | | **HAQA** | 81.10 | 68.59 | 72.07 | 37.20 | 53.91 | 80.37 | 52.42 | 41.37 | 65.54 |
| | INT8 | Human | 79.39 | 66.42 | 71.29 | 36.00 | 52.24 | 78.41 | 52.35 | 40.70 | 63.96 |
| | | Local | 79.98 | 67.79 | 71.62 | 36.87 | 51.94 | 79.08 | 52.18 | 40.04 | 64.25 |
| | | Bayesian | 78.22 | 68.60 | 70.84 | 36.42 | 50.41 | 79.12 | 52.04 | 39.54 | 63.71 |
| | | Random | 76.56 | 68.80 | 71.27 | 36.37 | 50.03 | 78.53 | 52.28 | 39.78 | 63.52 |
| | | NSGA2 | 74.36 | 67.02 | 71.18 | 36.37 | 49.55 | 78.83 | 52.52 | 40.08 | 63.05 |
| | | **HAQA** | 81.15 | 70.39 | 72.40 | 37.80 | 53.61 | 80.07 | 52.64 | 42.00 | 65.57 |
| LLaMA3.2-3B | INT4 | Human | 77.44 | 64.26 | 68.84 | 33.80 | 45.40 | 75.70 | 49.45 | 36.20 | 60.57 |
| | | Local | 77.53 | 70.39 | 69.14 | 33.80 | 47.16 | 77.06 | 49.21 | 35.80 | 57.51 |
| | | Bayesian | 75.97 | 71.40 | 68.26 | 33.60 | 45.63 | 76.85 | 49.32 | 35.05 | 57.01 |
| | | Random | 74.41 | 71.20 | 68.84 | 33.40 | 45.20 | 76.46 | 49.41 | 35.54 | 56.81 |
| | | NSGA2 | 72.16 | 69.67 | 68.55 | 33.60 | 44.82 | 76.66 | 49.60 | 35.44 | 56.31 |
| | | **HAQA** | 78.32 | 71.84 | 69.72 | 35.20 | 48.92 | 78.22 | 49.81 | 37.01 | 58.63 |
| | INT8 | Human | 78.40 | 63.00 | 70.21 | 33.60 | 47.46 | 77.73 | 49.70 | 36.60 | 61.07 |
| | | Local | 77.93 | 70.69 | 69.34 | 34.30 | 47.31 | 77.31 | 49.56 | 35.90 | 57.79 |
| | | Bayesian | 76.17 | 71.50 | 68.56 | 33.85 | 45.78 | 77.35 | 49.42 | 35.40 | 57.25 |
| | | Random | 74.51 | 71.70 | 68.99 | 33.80 | 45.40 | 76.76 | 49.66 | 35.64 | 57.06 |
| | | NSGA2 | 72.31 | 69.92 | 68.90 | 33.80 | 44.92 | 77.06 | 49.90 | 35.94 | 56.59 |
| | | **HAQA** | 79.10 | 73.29 | 70.12 | 35.23 | 48.98 | 78.30 | 50.02 | 37.86 | 59.11 |
| LLaMA3-8B | INT4 | Human | 83.03 | 68.17 | 74.80 | 36.40 | 54.29 | 81.34 | 52.95 | 41.80 | 66.34 |
| | | Local | 83.68 | 68.59 | 74.32 | 38.62 | 53.80 | 81.07 | 52.72 | 41.63 | 66.75 |
| | | Bayesian | 82.12 | 69.60 | 73.44 | 38.42 | 52.27 | 80.86 | 52.83 | 40.89 | 66.25 |
| | | Random | 80.56 | 69.40 | 74.02 | 38.22 | 51.84 | 80.47 | 52.92 | 41.37 | 66.05 |
| | | NSGA2 | 78.31 | 67.87 | 73.73 | 38.42 | 51.46 | 80.67 | 53.11 | 41.27 | 65.55 |
| | | **HAQA** | 84.47 | 70.04 | 74.90 | 40.02 | 55.56 | 82.23 | 53.32 | 42.84 | 67.87 |
| | INT8 | Human | 83.30 | 67.64 | 72.16 | 36.20 | 55.66 | 82.61 | 60.80 | 42.10 | 66.60 |
| | | Local | 83.22 | 66.35 | 72.95 | 38.67 | 53.29 | 81.88 | 60.61 | 41.43 | 66.43 |
| | | Bayesian | 81.46 | 67.16 | 72.17 | 38.22 | 51.76 | 81.92 | 60.47 | 40.93 | 65.89 |
| | | Random | 79.80 | 67.36 | 72.60 | 38.17 | 51.38 | 81.33 | 60.71 | 41.17 | 65.70 |
| | | NSGA2 | 77.60 | 65.58 | 72.51 | 38.17 | 50.90 | 81.63 | 60.95 | 41.47 | 65.23 |
| | | **HAQA** | 84.39 | 68.95 | 73.73 | 39.60 | 54.96 | 82.87 | 61.07 | 43.39 | 67.75 |

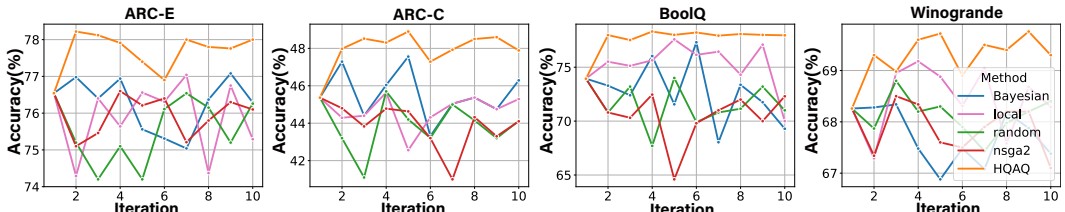

**Figure 4:** Convergence curve of HAQA and existing tuning approaches on multiple tasks.

practitioners Choi et al. (2018); Zhou et al. (2016). HAQA represents automated optimization via the agent, with a maximum of 10 tuning rounds, a constraint maintained throughout all experiments. The results show that HAQA consistently outperforms other methods across all quantization settings. Notably, in challenging scenarios such as w2a2, where the default setting fails to converge, HAQA achieves stable and superior performance.

We further evaluate HAQA on LLaMA models across different bit-widths and optimization methods, as detailed in Table 2. The results show that HAQA consistently outperforms human tuning and other methods. This improvement is particularly notable under INT4 settings, where HAQA significantly narrows the performance gap with FP16 models, demonstrating its robustness in low-bit quantization

**Table 3:** Kernel-Level Latency and HAQA Speedups.

| Kernel | Input Size | Default (μs) | HAQA (μs) | Speed-up |
|--------|-----------|-------------|-----------|----------|
| Softmax | [1024,1,32] | 3.45 | 2.57 | 1.34× |
|  | [1024,64,32] | 51.70 | 27.96 | 1.85× |
|  | [1024,128,32] | 98.15 | 52.87 | 1.86× |
| SiLU | [11008,1,1] | 6.29 | 5.11 | 1.23× |
|  | [11008,64,1] | 10.44 | 4.51 | 2.31× |
|  | [11008,128,1] | 31.02 | 19.71 | 1.57× |
| RMSNorm | [4096,1,1] | 10.19 | 8.61 | 1.18× |
|  | [4096,64,1] | 10.75 | 8.95 | 1.20× |
|  | [4096,128,1] | 11.11 | 9.10 | 1.22× |
| RoPE | [128,1,1] | 6.75 | 6.32 | 1.07× |
|  | [128,64,1] | 9.04 | 8.00 | 1.13× |
|  | [128,128,1] | 11.70 | 9.62 | 1.22× |
| MatMul | [2048,1,2048] | 16.49 | 12.24 | 1.35× |
|  | [2048,64,2048] | 52.29 | 36.86 | 1.42× |
|  | [2048,128,2048] | 63.20 | 38.85 | 1.63× |

**Table 4:** Model Throughput (Tokens/s) under Different Quantization.

| Model | FP16 | INT8 | INT4 |
|-------|------|------|------|
| openllama-3B | 5.11 | 5.25 | 4.95 |
| tinylama-1.1B | 11.17 | 11.23 | 10.43 |
| gpt2-large-774M | 13.41 | 13.20 | 12.29 |

**Table 5:** HAQA-Selected Configurations for `LLaMA2-13B`.

| Memory (GB) | FP16 | INT8 | INT4 |
|-------------|------|------|------|
| 4 | × | × | × |
| 12 | × | × | ✓ |
| 20 | × | ✓ | ✓ |
| 28 | ✓ | ✓ | ✓ |

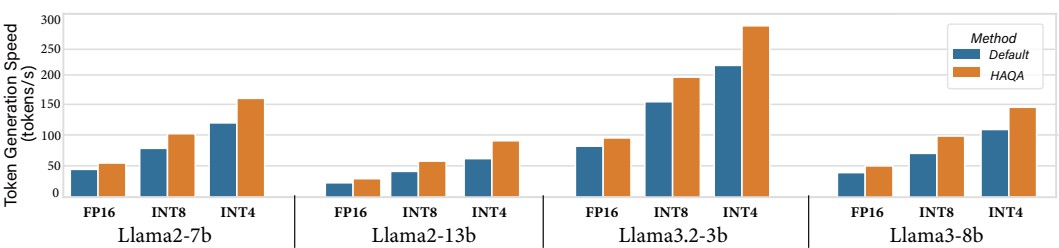

**Figure 5:** Token Generation Speed of Different Models Across Various Quantization Configurations (FP16, INT8, INT4).

scenarios. HAQA excels in systematically exploring the hyperparameter space and adapting to quantization challenges. Unlike manual tuning, which relies on fixed heuristics, it leverages iterative feedback and optimization strategies to refine hyperparameters, achieving better performance.

Additionally, We also report the convergence comparison results of HAQA and existing tuning approaches on multiple tasks using the LLaMA3.2-3b model (INT4). As illustrated in Figure 4. Beyond achieving higher accuracy, HAQA also converges more quickly and maintains greater stability throughout the optimization process, reinforcing its effectiveness as a robust solution for efficient model tuning under quantization constraints. This advantage stems from the agent's ability to leverage past tuning results and eliminate redundant trials, allowing it to refine hyperparameters more efficiently and accelerate convergence.

### 4.3 DEPLOYING ON HARDWARE PLATFORMS WITH EFFICIENT PERFORMANCE

To evaluate the inference efficiency of our approach on hardware platforms, we conduct three evaluations on an NVIDIA A6000 GPU: (1) analyzing the agent's ability to select appropriate quantization types while adhering to memory constraints, (2) evaluating optimization performance across various machine learning kernels commonly used in LLMs, such as softmax and matrix multiplication, and (3) measuring model execution latency to assess end-to-end performance improvements.

**Memory Constraint Evaluation:** Table 5 summarizes the quantization configurations selected by HAQA for deploying the LLaMA 2-13B model under different memory constraints. Given the available memory and hardware specifications, the agent automatically computes the memory requirements and selects a quantization type that satisfies the constraints. For instance, deploying the LLaMA 2-13B model with INT8 quantization requires 13 GB of memory. If only 12 GB is available, the agent rejects this configuration, as shown in Table 5.

**End-to-End Performance Analysis:** This section evaluates the HAQA's effectiveness in deploying end-to-end LLMs on hardware platforms. Figure 5 presents the inference speed (token generation rate) of various LlaMA models under different quantization types (FP16, INT8, INT4). As shown,

the agent-optimized implementation achieves significant speedups compared to the llama.cpp baseline ("Defaults"), with an acceleration of approximately $1.2\times$ to $1.5\times$. Furthermore, LLMs benefit from quantization as reduced bit-width computations lead to lower computational costs. For example, INT4 achieves higher generation speeds on the NVIDIA A6000 GPU due to its lower memory and compute requirements. Our agent-based optimization demonstrates superior performance under lower-bit quantization settings by effectively leveraging additional opportunities for execution optimization.

**Kernel-Level Performance Analysis:** Beyond end-to-end performance evaluation, Table 3 presents the execution latency of key computational kernels commonly used in LLMs. We compare our agent-optimized implementation with llama.cpp, demonstrating an acceleration of approximately $1.1\times$ to $2.3\times$ across various kernels with different computational scales. Notably, for the matrix multiplication kernel—one of the most computationally intensive operations in LLMs, typically accounting for 90% of inference runtime—our approach achieves a speedup of $1.35\times$ to $1.63\times$. This improvement stems from HAQA's ability to optimally configure execution parameters (e.g., tiling size, block parallelism) and adapt execution strategies (e.g., thread rescheduling, memory hierarchy optimization) to effectively reduce latency. These results underscore the effectiveness of our method in optimizing LLM inference on the target hardware platform.

## 4.4 ADAPTIVE QUANTIZATION STRATEGIES WITH HARDWARE-AWARE INTELLIGENCE

In our experiments with quantized models on different hardware platforms, HAQA provides counter-intuitive configuration suggestions, such as favoring INT8 over INT4 for improved performance on certain devices. For instance, on a OnePlus 11 mobile device powered by the Snapdragon 8 Gen 2 SoC—featuring an octa-core Kryo CPU and an Adreno 740 GPU—INT8 consistently outperformes INT4. This result is unexpected, as INT4, with its smaller bit-width, is generally assumed to yield faster evaluation due to reduced memory usage and computational demands. However, after extensive validation, HAQA's recommendations proved accurate: INT8 consistently achieves lower latency and higher inference speed, as shown in Table 4.

HAQA leverages its extensive knowledge and detailed hardware analysis to uncover the root causes of this phenomenon. It finds that on certain platforms, INT8 and INT4 computations benefit from specialized instruction sets or dedicated accelerators. For instance, the NVIDIA A6000 GPU, with Tensor Core Units, efficiently performs matrix multiplication and accumulation (MMA) using INT4 or INT8 while accumulating results in FP32, enabling high computational throughput for low-precision quantization. In contrast, on other hardware platforms such as mobile GPUs, direct inference using INT4 is not natively supported. To prevent data type overflow, INT4 and INT8 elements must first be converted to FP16, with subsequent accumulation also performed in FP16. Additionally, processing INT4 data often requires extra logistic operations (e.g., bitwise shifting, AND, OR) to reconstruct the original values. As a result, despite its theoretical efficiency, INT4 frequently fails to trigger optimized execution pathways on such hardware. Instead, INT4 operations are processed through general-purpose computation methods, negating their expected performance benefits over INT8. This capability highlights HAQA's effectiveness in identifying and validating nontrivial configuration choices, demonstrating its ability to optimize quantized models across diverse deployment environments while adapting to complex, hardware-specific constraints.

## 5 CONCLUSION

In conclusion, we present a user-friendly and highly efficient framework for deploying deep learning models, explicitly addressing numerous challenges for non-expert users. Our approach ensures reliable and consistent performance in diverse real-world applications while significantly reducing computational overhead, enabling deployment on resource-constrained edge devices. Looking ahead, our framework can uncover subtle issues and automatically optimize configurations often overlooked by experienced experts, rectifying them through adaptive, data-driven heuristics, ultimately fostering more innovative and transformative AI applications. Furthermore, this work lays a strong foundation for bridging the gap between state-of-the-art research and usability, fostering a future where AI becomes more accessible, transparent, and adaptable to diverse, rapidly evolving scenarios. Collectively, these advances pave the way for truly ubiquitous artificial intelligence, making sophisticated deep learning capabilities as accessible, reliable, and dependable as any off-the-shelf utility.

## 6 ETHICS STATEMENT

This work does not involve human subjects, personally identifiable data, or private information. We therefore believe this work does not raise ethical concerns.

## 7 REPRODUCIBILITY STATEMENT

We have made every effort to ensure the reproducibility of the results presented in this paper. A complete description of the algorithmic workflow is provided, and the Appendix includes full prompt examples used to obtain the experimental results. The models and baseline methods used in this study are based on publicly available open-source implementations to ensure consistency and reproducibility of the evaluation. We will release the complete code soon.

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

## A    LLM USAGE STATEMENT

Large language models (LLMs) were employed solely for polishing the manuscript, including refining clarity and checking grammatical errors and typos. LLMs did not contribute to the generation of substantive ideas, analyses, or results. All scientific content, interpretations, and conclusions are entirely the authors' own.

## B    MORE EXPERIMENTAL RESULTS

**Table 6:** Accuracy (%) (with standard deviation) comparison of different LLaMA models across various tasks and hyperparameter optimization methods using QLoRA.

| Model | Precision | Method | BoolQ | RTE | Winogrande | OpenBookQA | ARC-C | ARC-E | Hellaswag | MathQA | AVG |
|---|---|---|---|---|---|---|---|---|---|---|---|
| LLaMA2-7B | INT4 | Local | 76.46±0.62 | 67.15±0.58 | 68.95±0.47 | 35.60±0.32 | 46.48±0.51 | 76.86±0.45 | 51.26±0.29 | 38.63±0.34 | 61.99±0.57 |
| | | Bayesian | 74.90±0.49 | 68.16±0.60 | 68.07±0.44 | 35.40±0.36 | 44.95±0.53 | 76.65±0.41 | 51.37±0.33 | 37.88±0.39 | 61.49±0.55 |
| | | Random | 73.34±0.53 | 67.96±0.47 | 68.65±0.40 | 35.20±0.30 | 44.52±0.49 | 76.26±0.60 | 51.46±0.46 | 38.37±0.27 | 61.29±0.52 |
| | | NSGA2 | 71.09±0.45 | 66.43±0.51 | 68.36±0.35 | 35.40±0.42 | 44.14±0.47 | 76.46±0.39 | 51.65±0.58 | 38.27±0.61 | 60.79±0.44 |
| | | HAQA | 77.25±0.25 | 68.60±0.34 | 69.53±0.29 | 37.00±0.31 | 48.24±0.26 | 78.02±0.27 | 51.86±0.22 | 39.84±0.28 | 63.11±0.23 |
| | INT8 | Local | 79.13±0.56 | 68.88±0.43 | 69.46±0.61 | 37.67±0.38 | 48.03±0.33 | 78.01±0.58 | 51.44±0.42 | 39.17±0.49 | 62.90±0.44 |
| | | Bayesian | 77.37±0.37 | 69.69±0.55 | 68.68±0.32 | 37.22±0.47 | 46.50±0.52 | 78.05±0.30 | 51.30±0.59 | 38.67±0.41 | 62.36±0.27 |
| | | Random | 75.71±0.65 | 69.89±0.59 | 69.11±0.36 | 37.17±0.40 | 46.12±0.44 | 77.46±0.49 | 51.54±0.34 | 38.91±0.47 | 62.17±0.53 |
| | | NSGA2 | 73.51±0.46 | 68.11±0.38 | 69.02±0.52 | 37.17±0.62 | 45.64±0.48 | 77.76±0.33 | 51.78±0.55 | 39.21±0.57 | 61.70±0.30 |
| | | HAQA | 80.30±0.23 | 71.48±0.30 | 70.24±0.35 | 38.60±0.28 | 49.70±0.21 | 79.00±0.29 | 51.90±0.19 | 41.13±0.25 | 64.22±0.31 |
| LLaMA2-13B | INT4 | Local | 80.31±0.67 | 67.14±0.56 | 71.49±0.49 | 35.80±0.36 | 52.15±0.62 | 79.21±0.30 | 51.82±0.29 | 40.16±0.52 | 64.42±0.47 |
| | | Bayesian | 78.75±0.45 | 68.15±0.33 | 70.61±0.50 | 35.60±0.27 | 50.62±0.60 | 79.00±0.55 | 51.93±0.44 | 39.41±0.35 | 63.92±0.53 |
| | | Random | 77.19±0.51 | 67.95±0.40 | 71.19±0.54 | 35.40±0.47 | 50.19±0.48 | 78.61±0.63 | 52.02±0.37 | 39.90±0.31 | 63.72±0.57 |
| | | NSGA2 | 74.94±0.31 | 66.42±0.29 | 70.90±0.58 | 35.60±0.36 | 49.81±0.34 | 78.81±0.48 | 52.21±0.61 | 39.80±0.52 | 63.22±0.43 |
| | | HAQA | 81.10±0.28 | 68.59±0.21 | 72.07±0.31 | 37.20±0.23 | 53.91±0.19 | 80.37±0.25 | 52.42±0.22 | 41.37±0.26 | 65.54±0.33 |
| | INT8 | Local | 79.98±0.64 | 67.79±0.42 | 71.62±0.51 | 36.87±0.34 | 51.94±0.47 | 79.08±0.66 | 52.18±0.28 | 40.04±0.58 | 64.25±0.36 |
| | | Bayesian | 78.22±0.46 | 68.60±0.49 | 70.84±0.55 | 36.42±0.24 | 50.41±0.62 | 79.12±0.33 | 52.04±0.39 | 39.54±0.50 | 63.71±0.47 |
| | | Random | 76.56±0.57 | 68.80±0.52 | 71.27±0.46 | 36.37±0.39 | 50.03±0.41 | 78.53±0.48 | 52.28±0.68 | 39.78±0.32 | 63.52±0.53 |
| | | NSGA2 | 74.36±0.39 | 67.02±0.57 | 71.18±0.49 | 36.37±0.67 | 49.55±0.58 | 78.83±0.32 | 52.52±0.45 | 40.08±0.54 | 63.05±0.29 |
| | | HAQA | 81.15±0.24 | 70.39±0.23 | 72.40±0.39 | 37.80±0.25 | 53.61±0.22 | 80.07±0.18 | 52.64±0.30 | 42.00±0.21 | 65.57±0.27 |
| LLaMA3.2-3B | INT4 | Local | 77.53±0.54 | 70.39±0.44 | 69.14±0.35 | 33.80±0.48 | 47.16±0.61 | 77.06±0.27 | 49.21±0.33 | 35.80±0.56 | 57.51±0.52 |
| | | Bayesian | 75.97±0.42 | 71.40±0.53 | 68.26±0.47 | 33.60±0.30 | 45.63±0.58 | 76.85±0.40 | 49.32±0.49 | 35.05±0.36 | 57.01±0.44 |
| | | Random | 74.41±0.60 | 71.20±0.38 | 68.84±0.41 | 33.40±0.51 | 45.20±0.35 | 76.46±0.66 | 49.41±0.28 | 35.54±0.47 | 56.81±0.39 |
| | | NSGA2 | 72.16±0.45 | 69.67±0.62 | 68.55±0.27 | 33.60±0.49 | 44.82±0.52 | 76.66±0.38 | 49.60±0.59 | 35.44±0.32 | 56.31±0.58 |
| | | HAQA | 78.32±0.26 | 71.84±0.31 | 69.72±0.22 | 35.20±0.24 | 48.92±0.20 | 78.22±0.28 | 49.81±0.29 | 37.01±0.23 | 58.63±0.21 |
| | INT8 | Local | 77.93±0.63 | 70.69±0.55 | 69.34±0.36 | 34.30±0.69 | 47.31±0.48 | 77.31±0.29 | 49.56±0.51 | 35.90±0.43 | 57.79±0.65 |
| | | Bayesian | 76.17±0.35 | 71.50±0.47 | 68.56±0.58 | 33.85±0.39 | 45.78±0.60 | 77.35±0.32 | 49.42±0.46 | 35.40±0.66 | 57.25±0.41 |
| | | Random | 74.51±0.20 | 71.70±0.45 | 68.99±0.37 | 33.80±0.56 | 45.40±0.31 | 76.76±0.47 | 49.66±0.40 | 35.64±0.59 | 57.06±0.54 |
| | | NSGA2 | 72.31±0.33 | 69.92±0.52 | 68.90±0.29 | 33.80±0.61 | 44.92±0.54 | 77.06±0.28 | 49.90±0.42 | 35.94±0.65 | 56.59±0.55 |
| | | HAQA | 79.10±0.25 | 73.29±0.34 | 70.12±0.21 | 35.23±0.24 | 48.98±0.23 | 78.30±0.26 | 50.02±0.27 | 37.86±0.22 | 59.11±0.19 |
| LLaMA3.8-3B | INT4 | Local | 83.68±0.62 | 68.59±0.57 | 74.32±0.48 | 38.62±0.31 | 53.80±0.33 | 81.07±0.45 | 52.72±0.69 | 41.63±0.49 | 66.75±0.55 |
| | | Bayesian | 82.12±0.38 | 69.60±0.66 | 73.44±0.52 | 38.42±0.29 | 52.27±0.30 | 80.86±0.58 | 52.83±0.41 | 40.89±0.46 | 66.25±0.36 |
| | | Random | 80.56±0.61 | 69.40±0.59 | 74.02±0.32 | 38.22±0.44 | 51.84±0.68 | 80.47±0.36 | 52.92±0.47 | 41.37±0.53 | 66.05±0.57 |
| | | NSGA2 | 78.31±0.52 | 67.87±0.39 | 73.73±0.60 | 38.42±0.65 | 51.46±0.48 | 80.67±0.34 | 53.11±0.43 | 41.27±0.56 | 65.55±0.49 |
| | | HAQA | 84.47±0.28 | 70.04±0.24 | 74.90±0.29 | 40.02±0.22 | 55.56±0.26 | 82.23±0.27 | 53.32±0.25 | 42.84±0.34 | 67.87±0.21 |
| | INT8 | Local | 83.22±0.56 | 66.35±0.47 | 72.95±0.50 | 38.67±0.67 | 53.29±0.41 | 81.88±0.63 | 60.61±0.38 | 41.43±0.44 | 66.43±0.45 |
| | | Bayesian | 81.46±0.33 | 67.16±0.69 | 72.17±0.54 | 38.22±0.46 | 51.76±0.35 | 81.92±0.48 | 60.47±0.57 | 40.93±0.40 | 65.89±0.26 |
| | | Random | 79.80±0.68 | 67.36±0.36 | 72.60±0.47 | 38.17±0.39 | 51.38±0.52 | 81.33±0.34 | 60.71±0.49 | 41.17±0.59 | 65.70±0.53 |
| | | NSGA2 | 77.60±0.40 | 65.58±0.62 | 72.51±0.57 | 38.17±0.49 | 50.90±0.66 | 81.63±0.31 | 60.95±0.44 | 41.47±0.53 | 65.23±0.58 |
| | | HAQA | 84.39±0.27 | 68.95±0.25 | 73.73±0.23 | 39.60±0.24 | 54.96±0.22 | 82.87±0.28 | 61.07±0.21 | 43.39±0.32 | 67.75±0.19 |

## C    COST AND OVERHEAD OF HAQA

In our approach, we rely entirely on the off-the-shelf GPT-4-0613 API, eliminating the need to provision local GPUs, build custom serving infrastructure, or maintain on-premise hardware. The only runtime overhead is the API call itself: we measured an average round-trip latency of just 2.34 seconds per query, which is negligible. Cost-wise, under GPT-4's current list pricing, end-to-end optimization and deployment of two to three models consumes on the order of 150 K tokens, amounting to approximately $5 in total API fees. In other words, with only a few seconds of additional API latency per query and a total cost of approximately $5, our method unlocks substantial performance gains—offering a lightweight, highly cost-effective solution for the deplyment of LLMs.

## D    HYPERPARAMETER SEARCH SPACE

- **Llama-family models:**
    - Learning rate: $1 \times 10^{-5}$–$5 \times 10^{-4}$.
    - Batch size: 1–8.
    - Gradient accumulation steps: 4–32.

- Weight decay: $10^{-3}$–$10^{-1}$.
- Maximum training steps: 500–800.
- Maximum gradient norm: 0.1–1.0.
- LoRA rank and alpha: 8–64.
- Dropout probability: 0.0–0.3.

- **ResNet-style models:**
  - Learning rate: $1 \times 10^{-5}$–0.2.
  - Batch size: 32–256.
  - Weight decay: $10^{-6}$–$10^{-1}$.
  - Momentum: 0.5–0.99.
  - Total epochs: 10–24.

- **End-to-end deployment search:**
  - Loop transformations: six nest orders; tile sizes $8 \times 8$–$256 \times 256$.
  - Vectorization: SIMD widths 4–16 lanes.
  - Parallelization: thread, block, and grid dimensions 1–256.
  - Memory layout and prefetching: row-major or column-major; optional transposition; prefetch distances 0–16.

The Cartesian product of these parameters yields millions of configurations, making exhaustive tuning computationally prohibitive. Our work leverages an agent to strategically navigate this space, prioritizing promising configurations and dynamically pruning suboptimal ones.

# E  PROMPT SAMPLE

The following is a sample prompt used in our experiments. The specific code for quantization methods and hardware optimization varies by task and is quite lengthy, so we have omitted it.

mybox colback=gray!20, colframe=black, arc=5pt, boxrule=1pt, left=5pt,right=5pt, top=10pt,bottom=10pt, breakable

**Static Prompt Sample for ResNet32**

You are assisting in optimizing the hyperparameters for QAT of ResNet32. Using [8-bit] Quantization The dataset is CIFAR10. Code is based on PyTorch. Below is the hyperparameter search space:

'learning_rate': The learning rate for the optimizer. Type: UniformFloat,
Range: [1e-05, 0.2], Default: 0.01, Log scale.
'batch_size': The number of samples per batch of input data. Type: UniformInteger.
Range: [32, 256], Default: 128, Log scale.
'weight_decay': The L2 regularization coefficient. Type: UniformFloat.
Range: [1e-06, 0.1], Default: 5e-4, Log scale.
'momentum': The momentum for the SGD optimizer. Type: UniformFloat.
Range: [0.5, 0.99], Default: 0.9.
'num_epochs': The number of training epochs. Type: UniformInteger.
Range: [8, 12], Default: 12.

The number of epochs is relatively low because QAT training is performed on a pretrained model. You will receive accuracy results after each attempt. The goal is to find a configuration that minimizes the error rate within the given budget. If the loss remains unchanged, explore different parts of the search space. You should provide only \*\*one set of configurations per iteration\*\*. Once I provide the training results, you will return an optimized configuration.
Before making a decision, always generate a reasoning step (Thought) to analyze the current context, considering previous results and constraints. Then, take an appropriate action (Action) based on your reasoning. After the action, observe (Observation) the outcomes we feedback to you and adjust your approach accordingly.Identify missing information, potential errors, and formulate a strategy before taking any action. Each trial's configuration and results should be taken into account for a \*\*comprehensive\*\* analysis of the optimization process. Please review the history and consider your

next steps before proceeding.
**Make sure that all hyperparameters remain within the defined range**. ResNet32 is trained based on a **pretrained model** For the **first round**, it is recommended to use the **default parameters** for training.

Please provide the configuration in **JSON format**. For example:
{"learning_rate": x,
"batch_size": y,
"weight_decay": z,
"momentum": w,
"num_epochs": v}

The QAT Method is Dorefa, code are as follows: def DorefaW(w, bit, percent=0.01): xxx
Model structure: def q_resnet32(depth=32, dataset="cifar10"): xxx
Fine-tunig Code: xxx

**Complete Prompt Sample for ResNet32** { "role": "system",
"content": "You are an expert assistant specialized in optimizing hyperparameters for neural networks. Your goal is to help improve the performance of neural networks by providing optimized hyperparameter configurations." },
{ "role": "user",
"content": **[Static Prompt]** }
{ "role": "user",
"content": "Note that there are 7 rounds left. Please try to make effective attempts.
The current configuration is:{ "learning_rate": 0.005, "batch_size": 160, "weight_decay": 0.0007, "momentum": 0.9, "num_epochs": 12} The result based on this configuration: Verification accuracy: 0.8966. List of recent training losses(avg loss per epoch): [..] Please optimize and provide a set of optimized configurations." }
{ "role": "assistant",
"content": "From the training loss and accuracy list you provided, the performance of the model seems to be steadily improving, which shows that our current configuration is already quite good. However, we can also slightly increase the weight decay to further control possible overfitting. While the learning rate continues to decrease, we can try to make a little fine-tuning on the batch size. The suggested new CONFIG is as follows:{"learning_rate": 0.004, "batch_size": 170, "weight_decay": 0.0009, "momentum": 0.9, "num_epochs": 12} This time we still try to keep the model trained at a larger generalization error, while reducing the learning rate for fine-grained optimization." }
{ "role": "user",
"content": "Note that there are 6 rounds left, please try to make effective attempts. Finishing tasks with interleaving Thought, Action, Observation steps.
The current configuration is: { "learning_rate": 0.004,"batch_size": 170,"weight_decay": 0.0009,"momentum": 0.9,"num_epochs": 12}Results based on this configuration:Verification accuracy: 0.8966. List of recent training losses(avg loss per epoch): [..] Please optimize and provide a set of optimized configurations." }
{ "role": "assistant",
"content": "It seems that the last configuration did not bring the expected performance improvement, and the accuracy remained the same. Considering that our learning rate and weight decay are already relatively low, our next step may be to increase them appropriately, roll back the previous more aggressive optimization, and try to find better weights to improve the performance of the model. Here is my recommended configuration:{ "learning_rate": 0.006,"batch_size": 160,"weight_decay": 0.0007,"momentum": 0.9,"num_epochs": 12}. This time we increased the learning rate and weight decay, and slightly reduced the batch size to get more frequent parameter updates. Hopefully, this change will allow the model to find new features that will help improve accuracy. " }
{ "role": "user",
"content": "Note that there are 5 rounds left, please try to make effective attemptsThe current configuration is:{ "learning_rate": 0.006,"batch_size": 160,"weight_decay": 0.0007,"momentum": 0.9,"num_epochs": 12} The result based on this configuration:Verification accuracy: 0.8932. List of recent training losses(avg loss per epoch): [..] Please optimize and provide a set of optimized configurations" }
{ "role": "assistant",
"content": "From the given configuration, we can see that the accuracy has dropped slightly. Since

we have adjusted the learning rate and changed to a smaller batch size, now is a good time to try adjusting the momentum. Momentum can affect the speed of the model during gradient descent. If the momentum is set too high, the model may miss the minimum point, but if it is set too low, the model may fall into a local minimum. Here is the new configuration I recommend: { "learning_rate": 0.005,"batch_size": 150,"weight_decay": 0.0005,"momentum": 0.88,"num_epochs": 12} .This time we reduced the learning rate and momentum, hoping to make the model more carefully optimized. And keep the weight decay and batch size the same as the configuration with the highest accuracy. " } { "role": "user",
"content": "Note that there are 4 rounds left, please try to make effective attemptsThe current configuration is:{"learning_rate": 0.005,"batch_size": 150,"weight_decay": 0.0005,"momentum": 0.88,"num_epochs": 12 }Results based on this configuration:Verification accuracy: 0.8963. List of recent training losses(avg loss per epoch): [..] Please optimize and provide a set of optimized configurations" }

### Static Prompt Sample for Llama2-7b

You are helping optimize the hyperparameters of both [Qlora][(We use [8-bit] quantization)] fine-tuning and deployment for Llama2-7b.
I plan to deploy the model on an Ubuntu machine equipped with an NVIDIA A6000 GPU fully supporting TensorRT.Here's more details about the hardware:
{"Architecture":"Ampere","CUDA Cores":"10752","Tensor Cores":"336(3rd Gen)","RT Cores":"84(2nd Gen)","FP16 Performance":"309 TFLOPS","INT8 Performance":"618 TFLOPS","INT4 Performance":"1236 TFLOPS"}.
The memory limit is 10GB,Please choose an appropriate quantization bit width that satisfies the memory limitations and achieves better performance on such hardware.

Fine-tuning : The fine-tuning dataset is alpaca(tatsu-lab/alpaca) There are multiple validation datasets, and the results of each will be fed back to you Training is performed using PyTorch. Here is our hyperparameter search space:
'learning_rate': Learning rate for the optimizer. Type: UniformFloat, Range: [1e-5, 1e-3], Default: 4e-4, Use logarithmic scale.
'per_device_train_batch_size': Batch size for per-device training. Type: UniformInteger, Range: [4, 16], Default: 8.
'gradient_accumulation_steps': Number of steps for gradient accumulation. Type: UniformInteger, Range: [4, 32], Default: 8.
'weight_decay': L2 regularization coefficient. Type: UniformFloat, Range: [0.001, 0.1], Default: 0.01, Use logarithmic scale.
'max_steps': Maximum number of steps for training. Type: UniformInteger, Range: [200, 1000], Default: 400.
'max_grad_norm': Maximum norm for gradient clipping. Type: UniformFloat, Range: [0.1, 1], Default: 0.3.
'lora_r': Rank parameter for LoRA. Type: UniformInteger, Range: [8, 64], Default: 16.
'lora_alpha': Alpha parameter for LoRA. Type: UniformInteger, Range: [4, 32], Default: 8.
'lora_dropout': Dropout probability for LoRA. Type: UniformFloat, Range: [0.0, 0.3], Default: 0.05.
'warmup_ratio': warmup_ratio. Type: UniformFloat, Range: [0.0, 0.08], Default: 0.03.
You will get the accuracy after each trial. The goal is to find the configuration that minimizes the error rate within a given budget. If the loss does not change, explore different parts of the search space. You provide a set of configurations at a time, and when I give you the training results, you return a set of optimized configurations. Note that the hyperparameters must follow the given ranges. The configuration and data of each experiment must be taken into account in order to analyze the optimization configuration more comprehensively. Please note that the hyperparameters must be within the range I set It is recommended to use default parameters for training in the first round Please provide the configuration in JSON format. For example:
{"learning_rate": x, "per_device_train_batch_size": y, "gradient_accumulation_steps": z, "weight_decay": w, "max_steps": v, "max_grad_norm": p, "lora_r": a, "lora_alpha": b, "lora_dropout": c, "warmup_ratio" : d }
QLora core code: xxx
Model structure: xxx

Fine-tuning code: xxx

Deployment: The LLaMA model consists of various CUDA kernels, including Softmax, Silu, MatMul and so on. Please optimize the execution configuration and implementations of these kernels. The specific kernel information, default execution configuration, and core kernel code will be provided. The deployment latency results will be fed back to you. Here are two ways to optimize these kernels, first is to find the optimial kernel execution parameters, including computation block size for parallelization, tiling size and loop unrolling. Another one is to adopt execution strategy, specifying memory hierarchy settings for optimized tensor placement (e.g., global, shared, or local memory) and thread scheduling to balance workload distribution. If optimizing the execution configuration requires changes to the kernel implementation, you should also modify the Kernel Code (if necessary) ensure the modified code is correct and functional. Please provide the execution configuration parameters and the modified kernel code in the following JSON format:
{ "griddim":[x,y,z], "blockdim":[a,b,c], "tiling size": a, "code changed": true, "code":xxx }
The first kernel to Optimize is [Softmax], this is the execution informations about this kernel:
{ "kernel":"softmax", "tensor type":"float32", "src0 tensor shape": [64,1,32,1], "src1 tensor shape": [64,32,1,1], "output tensor shape":[64,1,32,1], "default gridDim":[32,1,1], "default block-Dim":[64,1,1], "unroll size" : 2 "kernel code":xxx }
Before making a decision, always generate a reasoning step (Thought) to analyze the current context, considering previous results and constraints. Then, take an appropriate action (Action) based on your reasoning. After the action, observe (Observation) the outcomes we feedback to you and adjust your approach accordingly. Identify missing information, potential errors, and formulate a strategy before taking any action. The configuration and data of each experiment must be taken into account in order to analyze the optimization configuration more comprehensively. Please review the history and consider your next steps before proceeding.

**Complete Prompt Sample for Llama2-7b**

{ "role": "system",
"content": "You are an expert assistant specialized in optimizing hyperparameters for both fine-tuning and deploymrnt of a certain neural network. Your goal is to help improve the accuracy and inference speed of the network by providing optimized hyperparameter configurations and code." }
{ "role": "user",
"content": **[Static Prompt]** }
{ "role": "user",
"content": "Note that there are 8 rounds left, please try to make effective attempts
Fine-tuning:The current fine-tuing configuration is: {"learning_rate":0.0008,"per_device_train_batch_size":8, "gradient_accumulation_steps":12, "weight_decay":0.01, "max_steps":400, "max_grad_norm":0.3, "lora_r":24, "lora_alpha":16, "lora_dropout":0.1, "warmup_ratio":0.05}.
The result of each task based on this configuration: Evaluation Result:{'hellaswag': {'alias': 'hellaswag', 'acc,none': 0.501953125, 'acc_stderr,none': 0.015632515719023321, 'acc_norm,none': 0.6650390625, 'acc_norm_stderr,none': 0.014756482342127149}, 'winogrande': {'alias': 'winogrande', 'acc,none': 0.689453125, 'acc_stderr,none': 0.01446698852617082}, 'openbookqa': {'alias': 'openbookqa', 'acc,none': 0.334, 'acc_stderr,none': 0.02111349234774374, 'acc_norm,none': 0.448, 'acc_norm_stderr,none': 0.02226169729227014}, 'arc_challenge': {'alias': 'arc_challenge', 'acc,none': 0.44921875, 'acc_stderr,none': 0.015551801140637223, 'acc_norm,none': 0.4853515625, 'acc_norm_stderr,none': 0.015625924747729628}, 'arc_easy': {'alias': 'arc_easy', 'acc,none': 0.76171875, 'acc_stderr,none': 0.01332000663047184, 'acc_norm,none': 0.767578125, 'acc_norm_stderr,none': 0.013205716756587446}}Please optimize and provide a set of optimized configurations.
The current execution configuration is:
{ "griddim": [64,1,1], "blockdim": [128,1,1], "tiling size": 4, "unroll size": 4, "code changed": true, "code": "} The result based on this configuration: {Latency : 0.619 us}. Please optimize and provide a set of optimized configurations" }
{ "role": "assistant",
"content": " Fine-tuning: It seems the evaluation results have slightly decreased compared to the initial setup. Additionally, looking at the list of training losses, the model doesn't seem to have improved significantly from the perspective of minimizing the loss. This could be an indication

that our learning rate might have been too high and the model might be skipping over potential minima. The increased dropout and LoRa hyperparameters might be causing the model to underfit the training data. Let's modify the hyperparameters accordingly. Here's the new set of configurations:{"learning_rate": 6e-4, "per_device_train_batch_size": 12,"gradient_accumulation_steps": 16,"weigh_decay": 0.02,"max_steps": 500,"max_grad_norm": 0.35,"lora_r": 20,"lora_alpha": 12,"lora_dropout": 0.07,"warmup_ratio": 0.04}. The strategy here is to slightly reduce the learning rate, increase the batch size and gradient accumulation steps to provide stability during optimization, and slightly increase max steps to give more time for optimization. As for the lora parameters, the idea is to promote a balance between fitting complexity and keeping the model simple enough to not overfit the data.

Deployment:The last optimization significantly improved performance, reducing latency to 0.619 μs. To further enhance efficiency, this iteration will focus on reducing memory bandwidth bottlenecks, improving thread utilization, and optimizing warp communication. Key Optimizations for This Round:

1.Increase blockDim to 256 Threads:More threads per block help with better utilization of GPU compute resources.

2.Adjust gridDim to 128 : Ensures more SMs are occupied, balancing computational workload distribution.

3.Optimize Shared Memory Usage: Using a dedicated shared memory buffer to store intermediate values for better locality.

4. Improve Memory Access Patterns Align memory loads using float4 to reduce memory transactions. here is Optimized Execution Configuration:

{ "griddim": [128,1,1], "blockdim": [256,1,1], "tiling size": 8, "unroll size": 8, "code changed": true, "code": xxx } }

{ "role": "user", "content": "Note that there are 7 rounds left, please try to make effective attempts.

Fine-tuning: The current configuration is:

{"learning_rate": 0.0006,"per_device_train_batch_size":12,"gradient_accumulation_steps": 16,"weight_decay": 0.02,"max_steps": 500,"max_grad_norm": 0.35,"lora_r": 20,"lora_alpha": 12,"lora_dropout": 0.07,"warmup_ratio": 0.04}.

The result based on this configuration: Evaluation Result:{'hellaswag': {'alias': 'hellaswag', 'acc,none': 0.544921875, 'acc_stderr,none': 0.015569414622825201, 'acc_norm,none': 0.6787109375, 'acc_norm_stderr,none': 0.014599990653924855}, 'winogrande': {'alias': 'winogrande', 'acc,none': 0.6953125, 'acc_stderr,none': 0.01439062049749023}, 'openbookqa':{'alias': 'openbookqa', 'acc,none': 0.37, 'acc_stderr,none': 0.021613289165165788, 'acc_norm,none': 0.482, 'acc_norm_stderr,none': 0.02236856511738799}, 'arc_challenge': {'alias': 'arc_challenge', 'acc,none': 0.4658203125, 'acc_stderr,none': 0.015596066531768368, 'acc_norm,none': 0.4775390625, 'acc_norm_stderr,none': 0.015616853889383952}, 'arc_easy': {'alias': 'arc_easy', 'acc,none': 0.76953125, 'acc_stderr,none': 0.013166833286747963, 'acc_norm,none': 0.7109375, 'acc_norm_stderr,none': 0.014173391116169621}} Please optimize and provide a set of optimized configurations.

Deployment: The current configuration is:

{ "griddim": [128,1,1], "blockdim": [256,1,1], "tiling size": 8, "unroll size": 8, "code changed": true, "code": xxx }

The result based on this configuration: {Latency: 0.630 us}, lease optimize and provide a set of optimized configurations." }

{ "role": "assistant", "content": "

Fine-tuning:The evaluation results show some improvement over the previous configuration. Specifically, there is an increase in accuracy in the 'hellaswag' and 'openbookqa' tasks. However, the 'arc_challenge' and 'arc_easy' tasks show a marginal decrease. The sequence of training losses does not indicate overfitting — it seems to plateau toward the end but does not increase significantly. This suggests that our model could benefit from additional expressiveness, either through longer training or larger rank and alpha parameters. For the next rounds, we will increase the 'max_steps' parameter to allow for more training. We'll also slightly increase the 'lora_r' and 'lora_alpha' parameters to increase the complexity of the learned representations. In addition, we will tweak the 'learning_rate' and 'per_device_train_batch_size' to facilitate quicker convergence.Here is the new set of configurations:{"learning_rate": 5e4,"per_device_train_batch_size":

16,"gradient_accumulation_steps": 16,"weight_decay": 0.015,"max_steps": 600,"max_grad_norm": 0.4,"lora_r": 24,"lora_alpha": 16,"lora_dropout": 0.07,"warmup_ratio": 0.04}. Please run the training with these parameters and let me know the results.

Deployment:The last optimization resulted in a slight regression in latency (0.630 μs), likely due to higher register pressure and shared memory contention from increasing the tiling and unroll sizes. To improve performance, this iteration will optimize register usage, reduce shared memory contention, and improve memory coalescing.Key Optimizations for This Round:
1.Reduce Block Size to 128 Threads: The previous increase to 256 threads may have caused excessive register pressure, reducing occupancy.
2.Reduce Unrolling Factor to 4 : The previous unrolling of 8 may have caused excessive register spills. Unrolling back to 4 will balance ILP (instruction-level parallelism) and register pressure.
3.Optimize Shared Memory Access: Reintroduce double-buffering for more efficient memory access.
4.Improve Memory Coalescing : Align loads with vectorized memory access using float4 where applicable.
here is optimized execution configuration:
{ "griddim": [128,1,1], "blockdim": [128,1,1], "tiling size": 4, "unroll size": 4, "code changed": true, "code": xxx } " }

## F  ADAPTIVE QUANTIZATION STRATEGIES FOUND BY HAQA

{ "role": "user",
"content": "I am running the model on a Oneplus 11 mobile device equipped with Snapdragon 8 Gen 2 SoC, featuring an octa-core Kryo CPU and Qualcomm Adreno 740 GPU with 16GB memory.Here's more details about the hardware: "Architecture": "Adreno (Snapdragon 8 Gen 2)", "ALUs (Shader Cores)": "768", "Tensor Cores": "Not Available (Uses AI Accelerators)", "RT Cores": "Supported (Hardware-accelerated Ray Tracing)", "FP16 Performance": "Up to 8 TFLOPS (Estimated)", "INT8 Performance": "Not Officially Disclosed ( 4-6 TOPS AI Performance)", "INT4 Performance": "Not Supported Natively (Emulated via INT8)"  The memory limit is 10GB,Please choose an appropriate quantization bit width that satisfies the memory limitations and achieves better performance on such hardware." }
{ "role": "assistant",
"content": "For your OnePlus 11 with Snapdragon 8 Gen 2 and Adreno 740 GPU, the best quantization choice is INT8. This is because Adreno 740 fully supports INT8 acceleration, providing a good balance between speed, accuracy, and memory efficiency.If accuracy is a priority and memory usage is less of a concern, you can use FP16, but it will consume more memory.  INT4 is not recommended since Adreno 740 does not natively support it, meaning it would require emulation, leading to slower performance." }

