# OpenReview forum: "From Bits to Chips: An LLM-based Hardware-Aware Quantization Agent for Streamlined Deployment of LLMs"
_ICLR.cc/2026/Conference — Submitted to ICLR 2026_

### Official Review · Reviewer_LR7t · 2025-10-24

**Soundness:** 2
**Presentation:** 3
**Contribution:** 2
**Rating:** 4
**Confidence:** 3

**Summary:**

The paper introduces HAQA, a framework that integrates LLM within a quantization and deployment pipeline to search for optimal hyperparameter settings. LLM proposes different configurations including bit-widths, learning rate, and hardware parameters in an iterative fashion, taking into account past runs. Authors report accuracy results on Resnet and llama models and compare results with A6000 and Snapdragon 8 Gen 2 hardware.

**Strengths:**

- The work introduces a novel idea. Leveraging LLMs to find quantization params seems like a timely idea.
- Clear motivation and writing.

**Weaknesses:**

- The proposed method has no mathematical foundation.
- The underlying usecase is fairly limited. While it’s largely useful for non-experts to leverage LLMs and optimize these model parameters, the process itself can be orders of magnitude more expensive in terms of compute as compared to traditional methods. While the cost of leveraging ChatGPT API calls is fairly cheap, in reality, they can be far more computationally intensive. It is not clear how LLM based optimization can be better than existing methods like grid search, bayesian optimization, etc. without any convergence guarantees.
- The framework may not perform well for new models and any new hardware intrinsics where the foundation model lacks knowledge of.
- Quantization generally involves params like scaling factor type, min/max range, etc and training includes params like learning rate, batch size, gradient accumulation steps, LoRA rank/dropout, etc. It is not clear whether and to what extent LLM based tuning helped with these methods. Existing methods leverage gradient-based information to guide this decision-making.

**Questions:**

- How sensitive are results to prompt wording and random seed?
- Did you evaluate failure cases (hallucinated configs, invalid bit-widths)?
- Could the same gains be achieved by a simple rule-based search conditioned on accuracy-latency feedback?

---

### Official Review · Reviewer_ySir · 2025-10-28

**Soundness:** 3
**Presentation:** 3
**Contribution:** 3
**Rating:** 6
**Confidence:** 3

**Summary:**

This paper proposes HAQA, a hardware-aware quantization agent framework based on large language models (LLMs), to automate the quantization tuning and hardware deployment process for large language models. By designing static and dynamic hints and leveraging the inference capabilities of LLMs, this framework jointly optimizes quantization training hyperparameters and hardware deployment configurations, significantly improving model inference speed and accuracy on resource-constrained devices. Experiments demonstrate that HAQA outperforms manual tuning and traditional optimization methods across multiple models and hardware platforms.

**Strengths:**

Highly Innovative: This approach utilizes LLM as an automated agent for the joint optimization of quantized training and hardware deployment, demonstrating its originality.

Highly Practical: This approach significantly reduces the difficulty of deploying quantized models for non-expert users, improving deployment efficiency and model performance.

Extensively Experimented: This approach has been extensively validated across multiple models, datasets, and hardware platforms, yielding compelling results.

Highly Hardware-Aware: This approach can detect counterintuitive configurations, demonstrating a deep understanding of hardware characteristics.

Low Cost: This approach can be optimized with only a few API calls , demonstrating excellent scalability.

**Weaknesses:**

Reliance on closed-source models: The experiments are based entirely on the GPT-4 API and lack validation against open-source LLMs, potentially impacting the generalizability and reproducibility of the method.

Lack of in-depth comparisons: While comparisons were conducted with various optimization methods, they were not fully compared with the latest hardware-aware quantization methods.

Complex prompt design: The design of static and dynamic prompts is complex, requiring users to understand and modify them.

Unverified generalizability: The experiments primarily focused on vision and language models, lacking validation of other modalities.

Failure cases not discussed: The potential failures that may occur during the agent optimization process and their handling mechanisms are not analyzed.

**Questions:**

Dependency on Proprietary LLMs:
The entire HAQA framework relies on the closed-source GPT-4 API. Have you experimented with open-source LLMs (e.g., Llama, Mistral) as the agent? If not, do you believe HAQA’s performance is generalizable across different LLMs, especially smaller or open-source ones?

Generalization to Other Domains:
The experiments focus on vision (ResNet) and language models (LLaMA). Can HAQA be applied to other model types, such as speech or multimodal models? Have you conducted any preliminary experiments in those areas?

---

### Official Review · Reviewer_C2Fi · 2025-10-30

**Soundness:** 2
**Presentation:** 2
**Contribution:** 2
**Rating:** 4
**Confidence:** 3

**Summary:**

This paper proposes a hardware-aware quantization agent centered on a large model, which unifies quantization fine-tuning and underlying kernel configuration into a single "feedback-driven" workflow, automatically performing joint parameter tuning in a vast and strongly coupled search space with hardware. The framework achieves accelerated results while maintaining or approaching accuracy across platforms through a closed loop of standardized prompts and runtime metrics, demonstrating the end-to-end, transferable idea and feasibility of automated deployment.

**Strengths:**

1. Closed-loop optimization driven by large model agents incorporates hardware information and operational feedback into decision-making, transcending the traditional fragmented process of "quantification first, then deployment".

2.It can adaptively select configurations based on the characteristics of different devices and backends, revealing and leveraging the non-intuitive optimality brought about by platform differences.

3.The search space, constraints and action templates are clear, with logs and boundary checks. The optimization process is controllable and auditable, making it easy to expand to new hardware or new operators.

**Weaknesses:**

1. The optimization criteria and convergence conditions for multi-objective compromises (accuracy, delay, throughput, and video memory) are not clearly described. The agent mainly relies on static/dynamic prompts and heuristic loops, and is easily affected by prompt design and model preferences. It is recommended to provide explicit multi-objective optimization objectives and constraints, and report failure modes and safety boundaries.

2. Lack of equal budget comparisons with standard automatic tuning /HPO baselines; Please provide the convergence curve and the optimal value distribution under the same number of parameters, training steps, data quota and evaluation calls, and report the variance and significance of multiple repetitions.

2. The key components of the agent workflow are not dissolved item by item. It is recommended to provide a robustness assessment under grid ablation and noise injection.

**Questions:**

1. Can you provide some intuitions or examples of what the better hyper-parameters are? And how does the agent find these parameters?

2. LLaMA/Alpaca results are unclear. While Llama models are mentioned trained on Alpaca dataset, which results in the table use QLoRA fine-tuning and which are quantization-only?

2. The paper only reports API costs and overhead. However, does the LLM-based agent need to run more trials or evaluations to reach the current performance than other search methods?

3. If so, would this make comparisons unfair? The added experiments and time increase should also be counted toward the total overhead. Please report these details.

---

### Official Review · Reviewer_1qQf · 2025-10-31

**Soundness:** 3
**Presentation:** 3
**Contribution:** 2
**Rating:** 2
**Confidence:** 4

**Summary:**

This paper introduces a hardware-aware quantization agent (HAQA) designed to facilitate the deployment of quantization algorithms on target hardware through efficient hyperparameter tuning and the knowledge of the hardware platform. The authors claim that HAQA can automatically identify optimal settings for various hardware platforms and contribute to accelerating inference speed.

**Strengths:**

The authors address a practical problem: the difficulty of efficiently identifying optimal configurations for different quantization techniques across diverse hardware platforms due to the vast search space. They also validate the effectiveness of their framework through experiments on various models.

**Weaknesses:**

1. It is unclear why this framework is specifically tailored for quantization. The proposed strategy contains few quantization-specific components beyond the target problem, deploying quantization techniques. For instance, it addresses only a limited number of quantization methods, which (e.g., QLoRa) are outdated, raising doubts about HAQA’s efficiency for current state-of-the-art techniques.

2. Although the framework aims to find optimal parameters/configurations for quantization techniques, some parameters, such as optimal data format, depend on activations (input data distributions), which HAQA cannot currently handle. Also, this framework may be more suitable for quantization-aware training (QAT) or fine-tuning rather than post-training quantization (PTQ), which is more widely used, while its efficiency relative to human experts also remains uncertain.

3. The claimed effectiveness across diverse hardware platforms is questionable, as HAQA relies on knowledge from existing LLM models. For practical cases involving deployment on new accelerators with unfamiliar hardware components or dataflows, applying HAQA directly may be challenging.

4. The authors suggest that HAQA can handle hardware platforms with unknown or unintuitive characteristics, citing an example where INT8 outperforms INT4 on mobile GPUs. However, in practice, supported data formats are usually pre-known (also included in authors’ prompt) and easily obtainable, making the claimed advantage less significant. Additionally, the search space and evaluation time for such parameters are relatively small. A more convincing case is needed.

5. Deploying a new quantization technique on novel hardware often requires generating efficient kernels and compilers, which remains a major bottleneck or deployment and is difficult to achieve reliably through LLM-based methods. This paper does not address this software stack issue, focusing only on hyperparameter or pipeline tuning, which may have limited impact on overall hardware performance.

Minor: There appear to be numerical inconsistencies in Table 2 (LLaMA-3.2B–3B), where the average human results outperform HAQA, even though individual task results are lower.

**Questions:**

Which aspects of this framework are specifically targeted toward quantization? Many components appear applicable to broader domains, as the framework primarily operates as an LLM agent without fine-tuning or training from scratch. Please clarify why this method is particularly effective for quantization and what novel contributions it offers from that perspective, beyond prompt engineering or leveraging existing hardware knowledge.

Can this approach be extended to novel hardware platforms that are not recognized by existing generative AI APIs? The deployment of new quantization techniques often hinges on hardware-specific information, which should be carefully addressed. Existing platforms are relatively straightforward to handle due to available prior results, but it is unclear how HAQA would perform on entirely new hardware. Additionally, please elaborate on how HAQA could provide advantages in the development of the software stack, including kernel generation and compiler optimization.

---

### Meta-Review · Area_Chair_JPz4 · 2026-01-08

**Summary:**

The paper proposes an LLM-based hardware-aware quantization agent to automate quantization and deployment. Reviewers found the problem important and practically motivated, with encouraging empirical results. However, they raised concerns about limited technical novelty, unclear quantization-specific contributions, weak methodological grounding, and insufficient comparisons to established automatic tuning methods. Questions were also raised about generalization to new hardware and modern quantization techniques.

**Reviewer Concerns:**

Reviewers concerns were not addressed since no author response was provided.



Still Outstanding:

- Unclear what is fundamentally new or quantization-specific beyond an LLM-driven tuning loop

- Lack of equal-budget comparisons with standard HPO or rule-based methods

- Limited support for modern quantization techniques and activation-dependent parameters

- Unclear generalization to novel hardware and missing discussion of software stack issues

- Insufficient ablations, robustness analysis, and failure case discussion

**Reviewer Scores:**

Reviewer scores would likely remain the same or slightly decrease since no author response was provided.

---

### Decision · Program_Chairs · 2026-01-26

Reject